# Pulmonary Embolism after Vaccination with the COVID-19 Vaccine (Pfizer, BNT162b2): A Case Report

**DOI:** 10.3390/vaccines11061075

**Published:** 2023-06-07

**Authors:** Eun-Ju Kim, Seok-Ju Yoo

**Affiliations:** 1Department of Infectious Disease Control, Ulsan Metropolitan City Hall, Ulsan 44675, Republic of Korea; behappyju@naver.com; 2Department of Preventive Medicine, College of Medicine, Dongguk University, Gyeongju 38066, Republic of Korea

**Keywords:** COVID-19 vaccines, BNT162 vaccine, pulmonary embolism

## Abstract

Pulmonary embolism causes pulmonary vascular obstruction and damages circulation, leading to death in serious cases. Various cases of thrombosis have been reported as adverse reactions after vaccination with COVID-19 vaccines, and reliable studies on thrombosis with thrombocytopenia syndrome (TTS) have been confirmed, especially for viral vector vaccines. However, the association with mRNA vaccines has not been proven. We report a case of pulmonary embolism and deep vein thrombosis that occurred after using mRNA COVID-19 vaccines (BNT162b2).

## 1. Introduction

COVID-19 is a new viral respiratory infection that first emerged in China in December 2019 and has resulted in many deaths worldwide. As the COVID-19 pandemic continued, each country sought various strategies, and Korea focused on suppressing the spread of the virus and forming collective immunity through vaccination [1]. With the recent introduction of COVID-19 vaccination, thrombosis with thrombocytopenia syndrome (TTS) has been reported [2,3], and most of its mechanisms are related to viral vector vaccines. Recent cases of pulmonary embolism after Pfizer vaccination (BNT162b2) have also been reported in several countries, including the United States, Morocco, and Oman [4,5,6].

Pulmonary embolism occurs when a deep vein thrombus separates and embolizes into the pulmonary circulation, resulting in pulmonary vascular occlusion and gas exchange and circulation impairment. Rapid progression to myocardial ischemia increases the likelihood of hypotension, syncope, and sudden death. The major risk factors for thromboembolism include myocardial infarction, cerebral infarction, immobility, surgery, old age, and malignant tumors [7]. In a situation where vaccination is unavoidable, there are concerns from minor adverse events to serious adverse events, but the COVID-19 vaccine is recommended at the national and individual levels.

In South Korea, through a self-controlled case study (SCCS), the incidence rate ratio of the post-vaccination risk period to the control period for deep vein thrombosis and pulmonary embolism following vaccination with COVID-19 was calculated; a significant increase in the incidence rate after one dose of Pfizer vaccine (IRR 1.22, 95% confidence interval 1.06–1.40, *p*-value = 0.005) was confirmed. As these results are inconsistent with those of overseas studies, a detailed epidemiological and clinical causality assessment of deep vein thrombosis and pulmonary embolism after Pfizer vaccination will be conducted in the future [8]. This case report is the first case of pulmonary embolism after COVID-19 (Pfizer, New York City, NY, USA) vaccination in our city. We report a case of pulmonary embolism and deep vein thrombosis that occurred after vaccination with a COVID-19 vaccine (Pfizer), with the approval of the Institutional Review Board (IRB approval no. P01-202304-01-016). Additionally, we have informed consent for the publication of clinical data by the study participants.

## 2. Case Presentation

A 58-year-old man with no underlying disease visited the emergency room via ambulance because of dyspnea. There was no underlying disease, and cardiac arrest occurred at the time of his arrival at the emergency room. He was resuscitated after CPR. There were no abnormal findings on the coronary angiography performed immediately after visiting the hospital. It was an emergency and coronary angiogram that had to be performed before Computed Tomography. On chest and abdomen angio-CT, pulmonary thromboembolism was accompanied by thrombosis in both main PAs, the lobar and segmental PAs of the left upper lobe (LUL) and left lower lobe (LLL), and the lobar and segmental PAs of the right middle lobe (RML) and right lower lobe (RLL) (Figure 1).

Additional transthoracic echocardiography (ECG) showed a D-shaped left ventricle (LV), confirming an enlarged right ventricle, but there was no regional wall motion abnormality (RWMA). The cardiac ejection fraction was normal at 67% (EF = 67%), and mild pulmonary hypertension was confirmed. Deep venous thrombosis coagulation was additionally confirmed on L/E vein CT (Figure 2).

Blood coagulation tests were delayed with a prothrombin time of 15.7 s, an international normalized ratio of 1.39, and an activated partial thromboplastin time of 37.8 s, and thrombocytopenia was confirmed by a platelet count of 119,000 cells/UL. Not only platelets but also red blood cells and hemoglobin were decreased.

D-dimer content increased to 23.6 μg/mL, and CRP levels increased, but myocardial enzyme tests such as Troponin-I and Creatine Kinase MB (CK-MB) showed no abnormality. Antibody tests related to autoimmune diseases, such as rheumatoid factor, antinuclear antibodies, anti-beta 2 glycoprotein 1 antibody, antineutrophil cytoplasmic antibodies, serum complement (complement 3, 4, and 5), and anti-cardiolipin antibody, were all negative, with no symptoms related to autoimmune disease reported.

Additional medical history was taken that could cause a pulmonary embolism. With the cooperation of Drug Utilization Review on The National Health Insurance Service and Health Insurance Review & Assessment Service, the patient’s major underlying diseases (renal failure, cerebrovascular disease, ischemic heart disease, hypertensive disease, and diabetes, etc.) and family history (pulmonary embolism, aortic dissection, and Guillain–Barr syndrome, etc.) were confirmed, but there were no specific findings. There was a history of drinking and smoking through the admission note, but the frequency, intensity, and duration were not exactly confirmed. Additionally, the history of exercise was not checked.

There was an absence of factors that could cause pulmonary embolism in the underlying disease and hematological examination, and we confirmed that the first dose of the COVID-19 vaccine (Pfizer, BNT162b2) had been received seven days earlier. In this regard, pulmonary embolism occurring after vaccination with COVID-19 was considered. In addition, since there had been cases in which platelet factor 4 antibody (anti-PF4), confirmed in vaccine-induced prothrombotic immune thrombocytopenia (VIPIT), was positive in the same situation as this patient, a test was planned for this. However, such a test could not be conducted for this patient because he received an mRNA vaccine rather than a virus vector vaccine; he was already taking a thrombolytic agent, and it was difficult to cooperate with medical institutions. According to the guidelines of the Korea Disease Control and Prevention Agency, PF4 samples are collected when the attending physician of a medical institution suspects a case of TTS (Thrombocytopenia Syndrome). However, the physician did not suspect TTS. As a result, it seemed to be a case of suspected VIPIT, but according to the doctor’s opinion, the administration of a thrombolytic agent and an anticoagulant was proposed, which did not aggravate the thrombosis.

Thrombolytic drugs (actilyse 10 mg IV bolus and actilyse 90 mg IV intravenous infusion for 2 h) were first administered at the beginning of hospitalization, and low-molecular-weight heparin treatment (enoxaparin 60 mg bid) was administered for 3 days according to the aPTT results. After 7 days of hospitalization, the patient was discharged because the general condition and examination results were good. Upon discharge, an anticoagulant (Rivaroxaban 15 mg 1T bid) was prescribed for 3 weeks, followed by outpatient follow-up, with no recurrence reported to date.

## 3. Discussion

Pulmonary embolism and deep vein thrombosis have a high fatality rate and a higher incidence in the West than in the East, but are gradually increasing in the South Korean population. Pulmonary embolism is a disease affected by various factors (gender, age, lifestyle, and underlying disease) [8], but to date, no clear link between pulmonary embolism and COVID-19 vaccination has been demonstrated. Unlike mRNA vaccines, significant results have been obtained linking virus vector vaccine (ChAdOX1) with thrombocytopenia to vaccine-induced thrombotic thrombocytopenia (VITT), vaccine-induced prothrombotic immune thrombocytopenia (VIPIT), or thrombosis with thrombocytopenia syndrome (TTS) [3,9,10].

However, adverse events that occur after vaccination refer to all unintended symptoms that occur after vaccination and do not necessarily require causality with vaccination [11]. In other words, an accidental reaction may be misunderstood as an adverse reaction caused by a vaccine, and conversely, even if a disease develops due to vaccination, it is difficult to prove it.

In the Republic of Korea, continuous research has been conducted by collecting big data on pulmonary embolism and deep vein thrombosis without thrombocytopenia. In the primary analysis, a small increase in the consistent incidence of the total combination index and the post-vaccination risk period compared to the post-vaccination control period of individual diseases was detected in the Pfizer vaccine (incidence ratio 1.22; 95% confidence interval 1.04–1.40) [8]. As a result of conducting an additional detailed analysis, there were insufficient grounds to admit the connection with the COVID-19 vaccine [12] (Table 1).

In most related reports, there were different case reports by vaccine type (virus vector vaccine, mRNA vaccine, etc.), presence or absence of thrombocytopenia, and PF4 antibody test results (positive or negative), but the possibility that it was caused by the COVID-19 vaccine could not be ruled out. In the United States, a 65-year-old male with chronic hypertension and hyperlipidemia developed a pulmonary embolism and deep vein thrombosis 10 days after the second dose of the Pfizer vaccine. The PF4 antibody test was positive, with accompanying thrombocytopenia [4]. In Morocco, a 49-year-old male without an underlying disease was diagnosed with pulmonary embolism 19 days after the second dose of the Pfizer vaccine, but without thrombocytopenia [5]. In Oman, a 59-year-old woman with diabetes and osteoporosis was diagnosed with pulmonary embolism and deep vein thrombosis seven days after the first dose of the Pfizer vaccine, similarly without thrombocytopenia [6]. In addition, pulmonary embolism after mRNA vaccination has been reported in several countries (Table 2) [4,5,6,13,14,15,16].

Furthermore, there are several reports related to thrombosis after vaccination with COVID-19 [17,18,19,20]. In Korea, a 28-year-old male without underlying disease developed cerebral vein thrombosis 11 days after vaccination with AstraZeneca (viral vector) [18]. In Germany, cerebral vein thrombosis occurred in 4.8% of patients after vaccination with the Pfizer (mRNA) vaccine [21]. In the UK, an increase in venous thrombosis, such as deep vein thrombosis, was observed in the early stages of vaccination, but subsequent studies showed no relevance [22]. In addition, the possibility that viral vector vaccines can cause blood clots as a very rare side effect has been raised, but there is a report that Pfizer and Moderna, which are mRNA vaccines, are considered safe and should continue to be used [19]. In other words, although conflicting research results are being reported, individual cases should not be overlooked because vaccination is recommended for the nation and individual health. In other words, blood clots after vaccination travel through the blood vessels and cause cerebral vein thrombosis, portal vein thrombosis, and pulmonary vein thrombosis when the brain, liver, and lungs are blocked, respectively. Therefore, it is necessary to closely examine the causality of vaccines in all venous thromboses.

According to the Korea Disease Control and Prevention Agency (KDCA), thrombocytopenic thrombosis is caused by viral vector (AstraZeneca (Cambridge, UK), Janssen (New Brunswick, NJ, USA)) vaccines only. Symptoms appear within 1 to 42 days, and if the PF4 ELISA test result is positive, causality is recognized. Although it is difficult to admit the causality of cerebral venous sinus thrombosis, medical expenses are supported because it is suspected of being related to vaccines. As for venous thrombosis, medical expenses are similarly supported if symptoms develop within 1 to 28 days of using the viral vector (Janssen) vaccine. In the treatment guidelines, thrombocytopenic thrombosis is suspected when it is diagnosed with a high degree of temporal probability (symptoms occur within four to 42 days of vaccination), suspected symptoms (headache, edema of extremities, purpura, abnormal bleeding, etc.), decreased platelets (<150 × 10^3^/uL) and increased D-dimer concentrations (>2.0 mcg/mL), and a positive PF4 ELISA test result is considered a confirmed case [23]. To date, in South Korea, there have been reliable reports of blood clots and platelet reduction after COVID-19 vaccination, but there is insufficient evidence to recognize causality, such as detailed mechanisms of occurrence.

In this case study, the authors reported pulmonary embolism after vaccination with COVID-19 (Pfizer, New York City, NY, USA). Although it had a different aspect from the venous thrombosis currently suggested by KDCA, the possibility of its association with the vaccine cannot be ruled out, as there was a temporal correlation with vaccination, no underlying disease, no family history, and no high risk factor for pulmonary embolism. Through this case, the authors would like to emphasize that the association with the COVID-19 vaccine should be considered when clinical and radiological findings suggesting pulmonary embolism are found in patients without other risk factors. Although the causal relationship between the mRNA vaccine and thromboembolism has not yet been identified, it is hoped that these published cases will be accumulated and serve as basic data to contribute to evaluating the exact causal relationship between the mRNA vaccine and thromboembolism in the future.

## 4. Conclusions

Vaccination is important at the national level because it protects individual health and prevents disease by blocking the spread of infectious diseases. Because safety studies on vaccines were done quickly in the context of the COVID-19 pandemic, many adverse events are being reported, unlike other vaccinations. Although it is difficult to take an individual approach to complex and diverse adverse events, cases that cannot be excluded from relevance to vaccination should be reviewed from multiple angles. Regardless of the type of vaccine, various thromboembolism cases are continuously being reported at home and abroad, so the examination of the relationship between the COVID-19 vaccine and thrombosis should continue.

## Figures and Tables

**Figure 1 vaccines-11-01075-f001:**
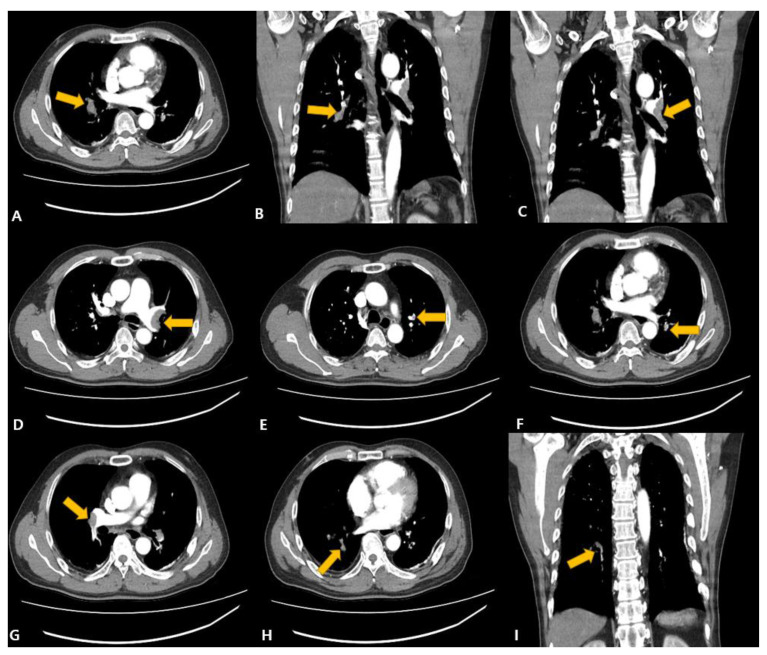
Chest aortic dissection angio-MDCT (Multi Detector Computed Tomography), (**A**) Right main pulmonary artery, (**B**) right middle lobe thromboembolism, (**C**,**D**) left main pulmonary artery, (**E**) left upper lobe pulmonary artery, (**F**) left lower lobe pulmonary artery, (**G**) right main pulmonary artery, (**H**,**I**) right lower lobe thromboembolism. Arrows point to suspected thrombosis sites.

**Figure 2 vaccines-11-01075-f002:**
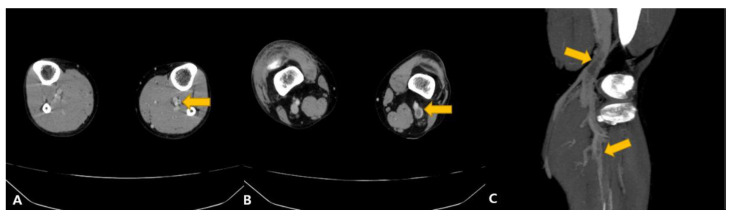
Lower Extremity venogram MDCT (Multi Detector Computed Tomography), (**A**) left calf veins, (**B**) left popliteal vein, (**C**) left popliteal vein, left calf veins. Arrows point to suspected thrombosis sites.

**Table 1 vaccines-11-01075-t001:** Summary of reported pulmonary embolism and deep vein thrombosis after COVID-19 vaccination in South Korea.

1. National Academy of Medicine of Korea [8], Presentation Date: 11 August 2022	
The Title of the Forum	COVID-19 Vaccine Safety Committee 4th Forum—3rd Study Results Presentation
Methods	• Design: Self-Controlled Case Series Minimize the involvement of confounding variables by comparing the incidence rate between the time when the effect of COVID-19 vaccination is estimated to occur (Risk interval) and the time when the effect is reduced (Post-vaccination Control interval).
Participants	6.5 million (Pfizer vaccine)
Outcome	The Pfizer vaccine showed a significant increase in the incidence rate after one vaccination (IRR 1.29, 95% confidence interval 1.05–1.58, *p*-value = 0.017).
Conclusion	• As a result of the analysis of thrombosis-related diseases such as deep vein thrombosis, when comparing the risk period after vaccination and the control period after vaccination, an increase in the incidence rate of all vaccines was not confirmed in the composite outcome, which gathered all related diseases. • However, for the Pfizer vaccine, a slight increase in the consistent incidence of the total binding index and the post-vaccination risk period compared to the post-vaccination control period of individual diseases was detected. • A detailed analysis is required, but this does not match the results of overseas studies, and there is a possibility of overdiagnosis due to concerns about adverse reactions. Therefore, a close epidemiological evaluation of deep vein thrombosis and pulmonary embolism after Pfizer vaccination will be conducted in the future.
**2. National Academy of Medicine of Korea [12], Presentation Date: 28 February 2023**	
The Title of the Forum	COVID-19 Vaccine Safety Research Center 3rd Forum
Methods	Design: Self-Controlled Risk Interval Study and Target Trial Emulation Study
Participants	2.4 million (Pfizer vaccine)
Outcome	There were no significant results.
Conclusion	• Previous analyses, such as deep vein thrombosis, did not confirm evidence to support causality. Therefore, in this reanalysis, the statistical association between domestic vaccination and deep vein thrombosis was evaluated using a strict case definition and two independent research methodologies. • In the self-controlled risk interval study and clinical trial simulation, no increase in the incidence of deep vein thrombosis was observed after vaccination. This is consistent with the results of some studies conducted abroad and a meta-analysis that synthesizes them.

**Table 2 vaccines-11-01075-t002:** Summary of the reported cases of Pulmonary Embolism after mRNA vaccination.

Author/ Country	Age/ Gender	Vaccine			Underlying Disease ^(1)^ and Adverse Events ^(2) †^	Blood ^(3)^ and Radiologic Observations ^(4) ǂ^	Findings or Conclusions
		Manufacturers	Dose	Time between Vaccination and Symptoms Onset			
1. Sangli, S. et al. [4]/USA	65/Male	Moderna (mRNA-1273)	2	10 days	(1) Chronic hypertension, hyperlipidemia (2) Bilateral lower-extremity discomfort, intermittent headaches, and dyspnea	(3) Platelet: 14 × 10^9^ cells/L (▼), D-dimer: 18.9 nmol/L (▲) (4) CT: large, bilateral, acute pulmonary emboli with right ventricular strain. Doppler: lower extremities revealed acute deep venous thromboses in both lower extremities.	This report presents the first report of VITT or TTS after a SARS-CoV-2 vaccine based on messenger RNA (mRNA) technology. Additionally, VITT or TTS, which was the cause of the adenovirus vector vaccine, may also occur in the mRNA vaccine.
2. Miri, C. et al. [5]/Morocco	49/Male	Pfizer (mRNA-BNT162b2)	2	7 days	(1) No medical history (2) Dyspnea	(3) Platelet: Normal, Increased CRP (▲), Elevated D-dimer (▲); (4) CT: proximal right pulmonary embolism.	This patient had no risk factors predisposing them to the development of acute venous thrombosis, in particular pulmonary embolism, and he tested negative for COVID-19 infection; however, the development of this thrombosis due to the mRNA-1273 vaccine is the most reasonable explanation.
3. Al-Maqbali, J.S. et al. [6]/Oman	59/Female	Pfizer (mRNA-BNT162b2)	1	7 days	(1) Type 2 diabetes mellitus, osteoarthritis (2) Chest pain, shortness of breath	(3) D-dimer: 24 mg/L (▲), Platelet: Normal; (4) Doppler: acute DVT involving the common femoral, superficial femoral, popliteal, posterior tibial, anterior tibial, and deep calf veins of the left lower limb; CT: saddle thrombus in the bifurcation of the pulmonary trunk and 40 extensive bilateral main pulmonary arteries emboli extending to the lobar segmental and subsegmental branches.	In the absence of an obvious explanation for the extensive DVT and bilateral PEs, and the proximity of COVID-19 vaccination, the authors believe that the patient’s presentation is probably related to a rare ADR of BNT162b2 mRNA COVID-19 (Pfizer-BioNTech).
4. Ishibashi Y. et al. [13]/Japan	74/Female	Moderna (mRNA-1273)	3	1 month	(1) Hypertension, dyslipidemia, and hypothyroidism (2) Pale, cold sweating, and hypoxic	(3) D-dimer 9.0 μg/mL (▲); Platelet: Normal (348 K/UL); (4) CT: thrombi in both pulmonary arteries.	This report is of a case of bilateral pulmonary embolism in a patient with no known risk factors for thrombotic events or previous episodes of APE, after the booster dose of the Moderna mRNA COVID-19 vaccine.
5. Cheong K.I. et al. [14]/Taiwan	70/Male	Moderna (mRNA-1273)	1	5 weeks	(1) Hypertension and an old cerebrovascular accident (2) Shortness of breath	(3) D-dimer: 4895 ng/mL (▲), Platelet: Normal; (4) Doppler: Left popliteal vein thrombosis.	COVID-19 vaccine may be the trigger for thrombosis in a patient with protein S deficiency.
6. Wiest N.E. et al. [15]/USA	66/Male	Moderna (mRNA-1273)	2	9 days	(1) Hypertension, hyper-lipidemia, and renal cell carcinoma (2) Right flank pain and right pleuritic chest pain	(3) D-dimer 3840 ng/mL (▲), Platelet: Normal (176 × 10^9^ cells/L); (4) CT: extensive multifocal pulmonary emboli involving both right and left lower lobe pulmonary arteries with evidence of right ventricular strain.	Thrombosis can occur after messenger RNA vaccination. Though commonly used for thrombosis, heparin may be ineffective. Non-heparin anticoagulants should be considered.
7. Ogunkoya J.O. et al. [16]/Africa	59/Male	Moderna (mRNA-1273)	3	1 month	(1) No medical history (2) Dyspnea, cough	(3) Platelet: Normal (175 × 10^9^ cells/L); (4) CT: pulmonary embolism of the right and left pulmonary arteries with features of possible early pulmonary hypertension.	This report describes the case of a 66-year-old male with no prior thromboembolic or hypercoagulable history who developed acute, bilateral pulmonary emboli promptly following his second Moderna SARS-CoV-2 immunization.

† Underlying Disease is past history, present illness and surgical history and Adverse Events is symptoms after COVID-19 vaccination. ‡ The following symbols indicate an increase (▲) or decrease (▼) from the normal range.

## Data Availability

All relevant data are presented in the manuscript.

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
