# Peer review of "Pulmonary Embolism after Vaccination with the COVID-19 Vaccine (Pfizer, BNT162b2): A Case Report"

_vaccines, 2023, doi:10.3390/vaccines11061075_

Round 1
Reviewer 1 Report
Dear Authors
Thank you for the work done here. You have illustrated the adverse effect of the vaccine encountered by the case studied here. In your discussion you have shown the comparison between different reported adverse reactions of different vaccines. I found you discussion very interesting. Even though the units of D dimmer were different but it’s understandable to stick to the source units
Author Response
Thank you for agreeing to the review.
Also, it is a case report that is not enough, but thank you for accepting it.
We tried to match the units for D-dimer, but we not match that the units were different when considering various units and decimal points at medical institutions. Thank you for your understanding.
Have a nice day and take care.
Thank you so much.
Reviewer 2 Report
I think the manuscript deals with an interesting aspect relating to the safety of the Covid-19 vaccination, however it is a case report, which does not and cannot provide innovative elements. The argument is already expressed in the literature and the clinical case does not add new inputs in this sense.
Furthermore, it is difficult in this case to associate a clinical aspect with vaccination, beyond the temporal criterion.
On the other hand, the patient in the study could not perform the anti-PF4 test and this further reduces the evaluation relating to the correlation with Covid-19 vaccination. The positivity of the test is, in fact, one of the few elements in the literature that correlates with the previous Covid-19 vaccination, as demonstrated by the same literature reported by the authors.
Apart from this, there are not many methodological elements to evaluate, because it is essentially the description of a clinical case, analogous to others present in the literature.
I think the clinical case describes well what happened but does not dwell much on the patient's past history, on his familiarity with possible pathologies, for example, to evaluate other possible causes of thromboembolism.
Author Response
Thank you for agreeing to the review.
Also, it is a case report that is not enough, but thank you for your comments and suggestions.
To evaluate other factors that may cause thromboembolism according to the reviewer's comments, the patient's underlying medical history, family history, and lifestyle history were added. Plase check the Line(80-88 & 184-185).
In addition, it was mentioned in the discussion section that this case report could contribute to the evaluation of the exact causal relationship between mRNA vaccine and thromboembolism(Line 188-192).
I look forward to seeing that what I revised was carried out well as your opinion. Have a nice day and take care.
Thank you so much.
Reviewer 3 Report
Thank you for the opportunity to review this case. A few comments:
- At the beginning of the case presentation, what does "through 119" mean?
- Did the patient have coronary angiogram before the CT?
- Line 87 - what does " and it was difficult to cooperate with medical institutions" mean?
- please use generic form of the drug Xarelto
- I'm not sure I understand why the two presentation announcements were included.
Nothing specific
Author Response
Thank you for agreeing to the review.
Also, it is a case report that is not enough, but thank you for your comments and suggestions.
According of your five comments, I revised like below.
1) At the beginning of the case presentation, what does "through 119" mean?
1-1) We changed that ‘though 119’ is ‘though ambulance'.
2) Did the patient have coronary angiogram before the CT?
2-2) It was an emergency and coronary angiogram had to be performed before Computed Tomography(Line 52-53).
3) Line 87 - what does " and it was difficult to cooperate with medical institutions" mean?
3-1) According to the guidelines of the Korea Disease Control and Prevention Agency, PF4 samples are collected when the attending physician of a medical institution suspects a case of TTS (Thrombocytopenia Syndrome). But the physician does not suspect for TTS(Line 98-101).
4) please use generic form of the drug Xarelto
4-1) We changed that ‘Xarelto’ is ‘Rivaroxaban’.
5) I'm not sure I understand why the two presentation announcements were included.
5-1) Though Two presentations the results are contradictory to what our case report is trying to say, we include them because the study conducted using big data in South Korea was also a meaningful result(Line 127-128). And It was written through the comments of editor.
I look forward to seeing that what I revised was carried out well as your opinion. Have a nice day and take care.
Thank you so much.
Round 2
Reviewer 2 Report
Analyzing the anamnestic aspects, it is noted that the patient in the case report had risk factors for cardiovascular disease, although the pulmonary embolism occurs temporally following the anti-Covid vaccination.